# CodonTransformer: a multispecies codon optimizer using context-aware neural networks

Adibvafa Fallahpour[1,2,6], Vincent Gureghian [3,4,6], Guillaume J. Filion [2] ✉, Ariel B. Lindner [3,4,5] ✉ & Amir Pandi [3,4,5] ✉

Degeneracy in the genetic code allows many possible DNA sequences to encode the same protein. Optimizing codon usage within a sequence to meet organism-specific preferences faces combinatorial explosion. Nevertheless, natural sequences optimized through evolution provide a rich source of data for machine learning algorithms to explore the underlying rules. Here, we introduce CodonTransformer, a multispecies deep learning model trained on over 1 million DNA-protein pairs from 164 organisms spanning all domains of life. The model demonstrates context-awareness thanks to its Transformers architecture and to our sequence representation strategy that combines organism, amino acid, and codon encodings. CodonTransformer generates host-specific DNA sequences with natural-like codon distribution profiles and with minimum negative cis-regulatory elements. This work introduces the strategy of Shared Token Representation and Encoding with Aligned Multi-masking (STREAM) and provides a codon optimization framework with a customizable open-access model and a user-friendly Google Colab interface.

The genetic code, a universal set of 64 three-nucleotide codons, instructs cellular protein production from genomes. The genetic code is degenerate, *i.e.*, most of the 20 amino acids can be encoded by multiple codons. These synonymous codons are used with diverse frequencies across organisms due to differences in the abundance of cellular tRNAs, protein folding regulations and evolutionary constraints[1–3]. The preferential selection among synonymous codons is called codon usage bias, a characteristic feature varying among species[4,5]. Thus, taking into account codon usage bias is required in designing DNA sequences for heterologous gene expression. The process of tailoring synonymous codons in DNA sequences to match the codon usage preference of a host organism is known as codon optimization[6–10]. The need for codon optimization has recently been increased by the drop in template-less DNA synthesis costs and the rapid advancements in de novo protein design[11–16].

Exploring the combinatorial space of synonymous codons arrangement is virtually impossible (~$10^{150}$ for a 300-amino acid protein with the average composition of UniProtKB/Swiss-Prot[17]). Traditional codon optimization approaches often rely on the selection of highly frequent codons, which can lead to resource depletion and protein aggregation, or on the insertion of rare codons in random locations that can cause protein misfolding and ribosome stalling[18,19]. These efforts should not only aim to enhance the targeted protein expression but also to avoid host toxicity due to tRNA pool perturbation. Evolutionary-inspired approaches such as codon harmonization[20,21] leverage the codon usage pattern in an original sequence, yet are limited to natural

[1]Vector Institute for Artificial Intelligence, Toronto, ON, Canada. [2]University of Toronto Scarborough; Department of Biological Science, Scarborough, ON, Canada. [3]Sorbonne Université, CNRS, ERL U1338 Inserm, Department of Computational, Quantitative and Synthetic Biology, Paris, France. [4]Sorbonne Université, CNRS, Inserm, Institut de Biologie Paris-Seine, Paris, France. [5]Sorbonne Université, CNRS, Université de Technologie de Compiègne, Inserm, Biofoundry Alliance Sorbonne Université, Paris, France. [6]These authors contributed equally: Adibvafa Fallahpour, Vincent Gureghian. ✉e-mail: guillaume.filion@utoronto.ca; ariel.lindner@inserm.fr; amir.pandi@inserm.fr

proteins and to designing cross-species DNA among organisms with similar translation mechanisms and dynamics.

Deep neural networks, with their capacity to learn complex patterns and relationships in data, offer a promising approach to decipher the language of codon usage and efficiently design DNA sequences. Deep learning models have been developed laying the foundation of this technology for codon optimization[18,22–24]. Yet the potential of these models can be expanded by addressing limitations posed by small training data focused only on a single host organism (*e.g., Escherichia coli*), or limited model accessibility, and lack of user-friendly interfaces. Importantly, organism-specific global and local codon patterns have not been addressed, representing a substantial aspect of codon optimization that can be implemented via neural network architectures and specialized sequence encoding.

Here, we present a codon optimization approach that harnesses the capacity of Transformer architecture to capture intrinsic sequence patterns. As the training data, we used a vast collection of ~1 million gene-protein pairs from 164 organisms ("Data availability") that not only increases the data size and learning universal rules but also enables species-specific DNA design via a single model. To address organism-specific context-awareness, we used a sequence representation strategy combining organism encoding with tokenized amino acid-codon pairs. Hence, we introduce the strategy of Shared Token Representation and Encoding with Aligned Multi-masking (STREAM). We develop CodonTransformer, a multispecies model that learns codon usage patterns across organisms and designs host-specific DNA sequences. The base (pretrained) model is available to the community for custom fine-tuning on a user-defined set of genes. In this work, we fine-tuned the model on the 10% genes with the highest codon similarity index (CSI)[25] for 15 genomes (including two chloroplasts). CodonTransformer generated DNA sequences with natural-like codon distributions while minimizing the number of negative cis-elements.

Along with the open-access base and fine-tuned models, we provide a comprehensive Python package to streamline the entire codon optimization workflow, including dataset processing, model training, and sequence evaluation. We also offer a Google Colab notebook with a user-friendly interface for designing codon-optimized sequences using CodonTransformer.

## Results

### A Transformer model with combined organism-amino acid-codon representation

Optimizing a coding sequence can be thought of as reverse-translating a protein sequence into a DNA sequence. It is therefore tempting to approach the problem from the point of view of machine translation, which typically follows one of two paradigms: In the Encoder-Decoder approach, the query is first encoded into the hidden state of a neural network and then decoded into the desired language[26,27]. In the Decoder-only approach, the query is used as a prompt to be completed with the translation in the desired language[28]. In both approaches, the translation is produced by an auto-regressive decoder, meaning that one token at a time is produced until the translation is completed. The auto-regressive paradigm may be problematic in the context of codon optimization: Choosing a codon in the 5′ part of the sequence may cause interference in the 3′ part of the sequence, however once a codon is chosen it cannot be removed. It is preferable to use a bi-directional strategy where the sequence is optimized uniformly. We therefore used an Encoder-only architecture and trained it with a masked language modeling (MLM) approach.

In MLM, parts of the sequence are hidden and the task of the algorithm is to recover the missing parts using the information from the parts that are available. This design is bidirectional; for instance it allows the user to optimize a region in 5′ while keeping the rest of the sequence constant, which is not possible with an Encoder-Decoder or a

Decoder-only architecture. Remains the problem of instructing the algorithm that it has to produce the DNA template of a given protein. For this, we designed a specialized alphabet and a tokenization scheme where a codon can either be clear or hidden. In this alphabet, the symbol A_GCC specifies an alanine residue produced with the codon GCC, and the symbol A_UNK specifies an alanine residue but does not specify the codon. The same logic is applied to all the codons, so that in effect we expanded the alphabet with 20 different MASK tokens, each indicating a different residue. During training, a fraction of the symbols are replaced with their masked version, and during inference, the input sequence uses the masked versions of all the symbols so that the algorithm can propose an optimized DNA encoding of the target protein.

The encoder model is based on the BigBird Transformer architecture[29], a variant of BERT[30] developed for long-sequence training through a block sparse attention mechanism (Fig. 1a). An essential requirement is to allow the algorithm to adapt the codon usage to the organism where the protein is expressed. This implies that during training, the algorithm must be aware of the source organism for a given DNA sequence and that during inference, users must be able to force the algorithm to use the context of their choice. We solved both issues by repurposing the token-type feature of Transformer models like BigBird. Token types are often used to distinguish interlocutors in text data (*e.g.*, question vs answer or user vs assistant) but they can be used to specify any type of context for string-like data. We therefore amplified the token type vocabulary so that every species has its own token type (Fig. 1b). This strategy allows our model to learn distinct codon preferences for each organism, associating specific codon usage patterns with their corresponding species. In addition, passing the token type as an argument allows users to optimize a DNA sequence in the species of their choice.

We trained the base model, which we named CodonTransformer, using ~1 million genes from 164 organisms ("Data availability"). The training set is a collection of genomes from all domains of life, *i.e.*, Bacteria, Archaea, and Eukarya that constitute 56.1%, 2.5% and 41.4% of sequences, respectively. This model can be either used directly for codon optimization across species or it can be fine-tuned on custom sets of DNA sequences to perform more tailored tasks. In this study, we fine-tuned CodonTransformer on the 10% genes with the highest CSI ("Methods") for 15 organisms: *Escherichia coli, Bacillus subtilis, Pseudomonas putida, Thermococcus barophilus, Saccharomyces cerevisiae, Chlamydomonas reinhardtii* and its chloroplast, *Arabidopsis thaliana, Nicotiana tabacum* and its chloroplast, *Caenorhabditis elegans, Danio rerio, Drosophila melanogaster, Mus musculus,* and *Homo sapiens* (Fig. 1c). The CSI[25] is derived from the codon adaptation index (CAI)[8] to quantify the similarity of codon usage between a sequence and the codon usage frequency table of an organism instead of an arbitrary reference set of highly expressed genes. It therefore provides a robust metric at the multispecies level and may constitute a superior predictor of expression level in higher eukaryotes[4,7,25,31].

### CodonTransformer learned codon usage across organisms

To assess the ability of the model to capture organism-specific codon preferences, we generated DNA sequences for all proteins encoded by the 15 genomes for which we performed fine-tuning (Fig. 1c). We then compared the sequences generated by the base and fine-tuned models to their natural counterparts (entire genome and top 10% CSI used for fine-tuning).

Sequences generated by the base model show a higher percentage of matching codons with their natural DNA counterparts than randomly selected codons without and with organismic preferences, uniform random choice (URC) and background frequency choice (BFC), respectively (Supplementary Fig. 1). Those sequences have a high CSI indicating that they follow for each organism the preference of codon usage (Fig. 2a, Supplementary Figs. 2–16). Obtaining higher

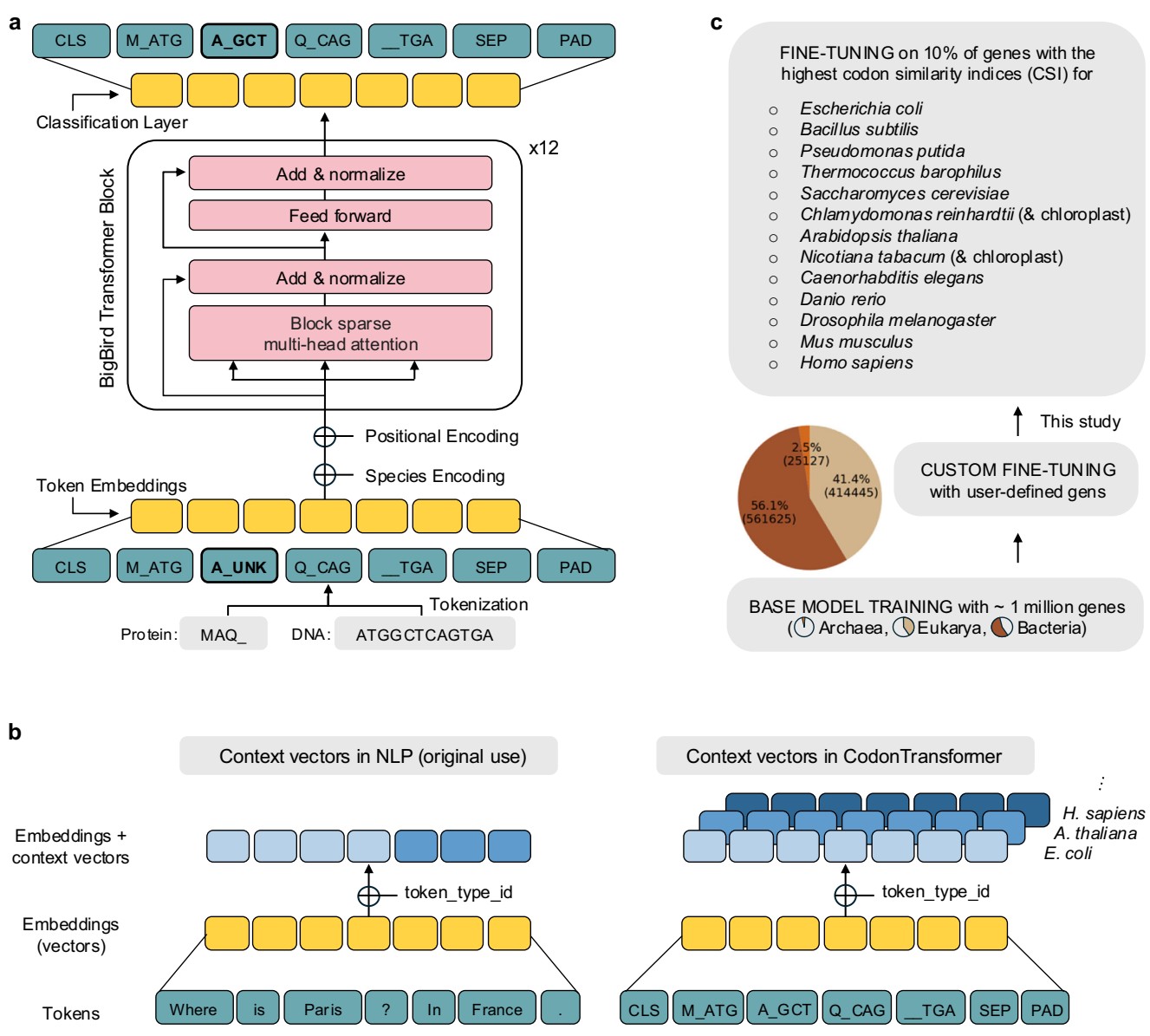

**Fig. 1 | CodonTransformer multispecies model with combined organism-amino acid-codon embedding. a** An encoder-only BigBird Transformer model trained by combined amino acid-codon tokens along with organism encoding for host-specific codon usage representation. **b** Schematic representation of the organism encoding strategy used in CodonTransformer using token_type_id, similar to contextualized vectors in natural language processing (NLP). **c** CodonTransformer was trained with ~1 million genes from 164 organisms across all domains of life and fine-tuned with highly expressed genes (top 10% codon usage index, CSI) of 13 organisms and two chloroplast genomes. CLS the start of sequence token, UNK general unknown token, SEP the end of sequence token, PAD padding token.

CSI than genomic sequences is expected because not every DNA sequence of a species is tuned for optimal expression, whereas CodonTransformer optimizes every sequence based on codon frequencies that it encounters during training. The base model generated sequences with higher CSI than the top 10% genomic CSI for all organisms except *S. cerevisiae*, *N. tabacum* and its chloroplast (Fig. 2a, Supplementary Figs. 2–16). When fine-tuning the model, Codon-Transformer generated sequences with lower CSI than the base model for all organisms except *S. cerevisiae*, *N. tabacum* and its chloroplast which showed an increase in the CSI better mimicking the top 10% genomic CSI on which they were fine-tuned (Fig. 2a, Supplementary Figs. 2–16). As expected, the clustering of CodonTransformer embeddings for organisms used for training resembles their overall phylogenetic distances (Supplementary Fig. 17).

Interestingly, all generated sequences have a GC content similar to their natural counterparts (Supplementary Figs. 2–16) further supporting that the model learned organism-specific characteristics. Finally, the tendency of the base model to pick rare codons, as measured by the Codon frequency distribution (CFD), representing the number of rare codons[18] (frequency <0.3) in a sequence, is very distinct across species. The influence of fine-tuning on CFD was striking for *P. putida, C. reinharditi* and its chloroplast where it increased the number of rare codons and for *N. tabacum* and its chloroplast for which it decreased the number of rare codons (Supplementary Figs. 4, 7, 8, 10, 11).

Although we developed CodonTransformer for sequence design tasks, it can also be used to predict the effect of synonymous mutations. To do this, the probability of mutant and wild-type codons can

be predicted by the model for every amino acid position. We used 62 synonymous mutations in *ccdA, i.e.,* the antitoxin component of the *E. coli ccdAB* toxin-antitoxin system, from a recent study[32] with their

respective experimental relative fitness and ribosome stalling values. Both base and fine-tune CodonTransformer models predicted the log-likelihood of the mutant codon significantly correlated with

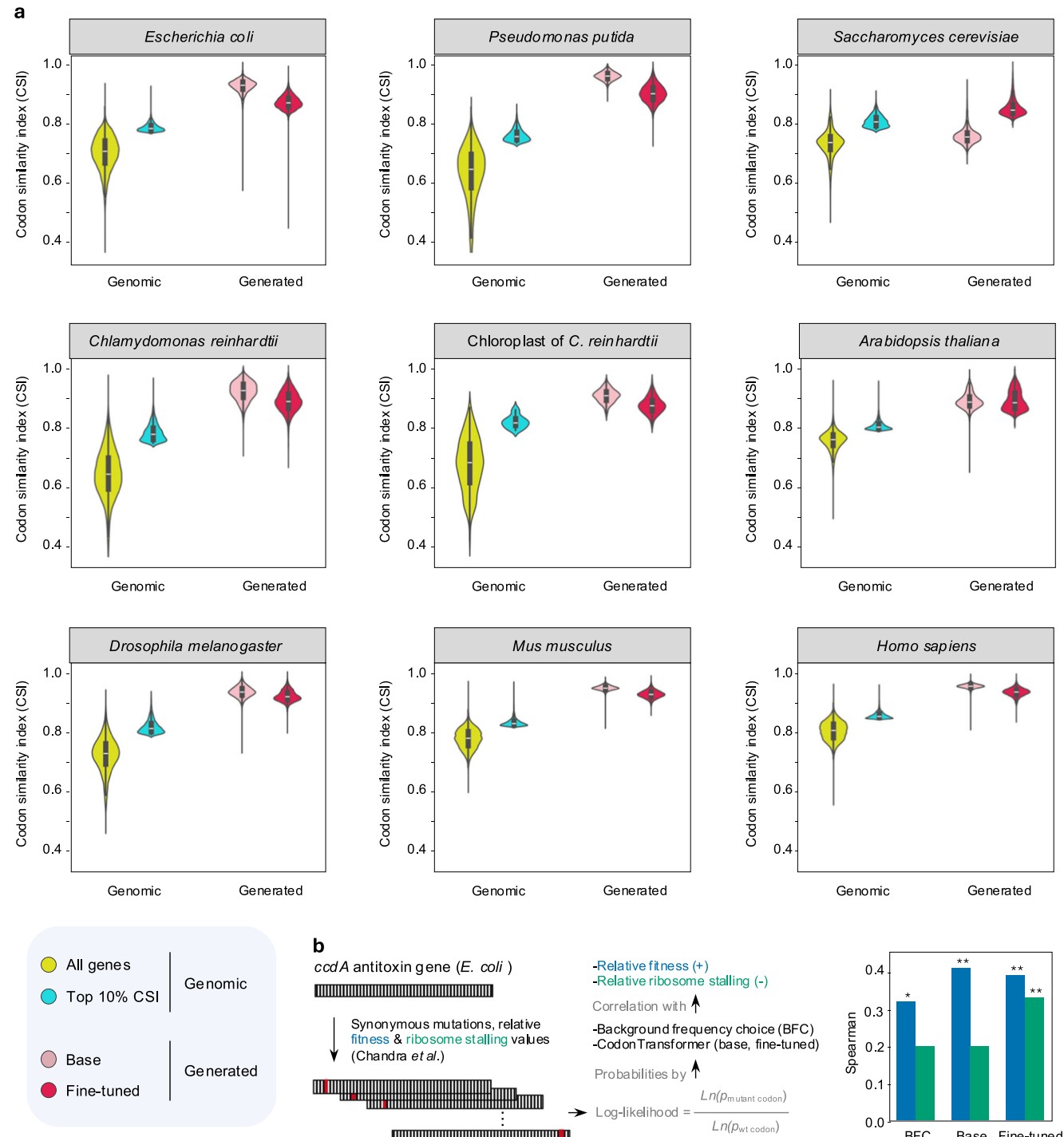

**Fig. 2 | CodonTransformer learned codon patterns across organisms. a** Codon usage index (CSI) for all and the top 10% CSI original genes (yellow and blue, respectively) and generated DNA sequences for all original proteins by Coron-Transformer (base and fine-tuned models, light and dark red, respectively) for 9 out of 15 genomes used for fine-tuning in this study. See Supplementary Figs. 2–16 for all 15 genomes and additional metrics of GC content codon and distribution frequency (CDF). **b** Synonymous mutations in the *E. coli* (K12 strain) *ccdA* antitoxin gene, from the ccdAB toxin-antitoxin system, were analyzed using Codon-Transformer (base and fine-tuned for *E. coli* K12 strain, with wild-type DNA as input to the model) and background frequency choice (BFC) models. The natural log of

the probability of mutant codons over wild-type codons was computed for 62 mutations from Chandra et al.[32], plotted against the natural log of experimental relative fitness, blue bars, (positive correlation) and relative ribosome stalling, green bars, (negative correlation, absolute values plotted). Two-sided Spearman correlation tests were used (for *n* = 62 mutations) to evaluate the models' performance with numerical p-values of, from left to right for the six bars, 0.0153 (*), 0.1232, 0.0015 (**), 0.1123, 0.0026 (**), 0.0094 (**). Raw data and source data for **a** and Supplementary Figs. 2–16 are available at https://zenodo.org/records/13262517 and for data underlying **b** is provided in the Source Data File.

experimental relative fitness in a zero-shot manner with the wild-type DNA sequence as input to the model (Fig. 2b). The same prediction task by BFC that selects codons based on organismic codon usage table, also resulted in significant correlations slightly less than CodonTransformers. However, among BFC, base and fine-tuned CodonTransformer, only the fine-tuned model showed a significant correlation between log-likelihood of mutation and relative ribosome stalling (Fig. 2b).

Altogether, these results demonstrate that the base model efficiently learned organism-specific codon usage preferences and that additional fine-tuning can further adjust the model outcomes.

## CodonTransformer generates DNA sequences with natural-like codon usage patterns

The strength of Transformers lies in their ability to capture long-range patterns in sequences, enabling CodonTransformer to generate DNA sequences with distributions of both low- and high-frequency codons. Codon distribution (and not only composition) plays an important role in the expression of both natural and heterologous genes by influencing the dynamics of translation in the host and the subsequent protein folding[1,33–40]. To quantify and visualize the codon usage pattern along a DNA sequence, %MinMax[41] provides a suitable metric[24] (Fig. 3a). %MinMax quantifies low- and high-frequency codons in a sliding window of 18 codons providing a local score throughout the sequence, with lower values indicating a higher presence of rare codons in the cluster[41].

For 5 model organisms (*E. coli*, *S. cerevisiae*, *A. thaliana*, *M. musculus*, and *H. sapiens*), we selected a set of 50 random genes among the top 10% CSI. We then compared their natural DNA sequence with the ones generated by CodonTransformer, both base and fine-tuned, and by Twist Bioscience, Genewiz, Integrated DNA Technologies (IDT), and ICOR (Improving Codon Optimization with Recurrent Neural Networks) which has been trained on and can codon optimize only for *E. coli*[18]. We first calculated the %MinMax profiles for all natural and codon-optimized sequences (the profile for one gene as representative of the dynamic time warping (DTW) distance for each organism is shown in Fig. 3b, and for all genes in the Source Data File). We then used the DTW algorithm ("Methods") to evaluate the distance between the %MinMax profiles of natural and generated sequences (Fig. 3c). Finally, for the 50 sequences of each organism, we computed the mean and standard deviation of the normalized DTW distances, that accommodate for differences in sequence length, to further compare the different tools (Fig. 3d, Supplementary Fig. 18).

The DTW distance is not significantly different between sequences generated by base and fine-tuned CodonTransformer, except for *E. coli* that the fine-tuned model showed a higher match to natural genes (*i.e.*, lower DTW) (Fig. 3c). Fine-tuned CodonTransformer performed as well or better than other models except in *A. thaliana* where Twist DTW distances were significantly lower. Overall, fine-tuned model and Twist generated sequences with lower normalized DTW distances indicative of more natural-like codon patterns (mean of 0.08 ± 0.05 and 0.07 ± 0.05, respectively) than other models (mean of 0.11 ± 0.05 for the base model, 0.14 ± 0.07 for IDT and 0.15 ± 0.09 for Genewiz) (Fig. 3d). ICOR performed well for *E. coli* sequences (0.08 ± 0.02).

Additionally, in order to assess our model capacity to capture potential RNA secondary structure constraints, we computed the minimum free energy for the corresponding RNA sequences based on the ViennaRNA package[42]. This energy was proportional to the sequence length (Supplementary Fig. 19a) and was normalized for it. Linear regressions between the normalized energy of generated sequences and natural ones showed that fine-tuned CodonTransformer and Genewiz better fitted natural energies than other models (Supplementary Fig. 19b). At the species level, both fine-tuned and Genewiz mimicked the separation between species observed for natural sequences, with both *S. cerevisiae* and *A. thaliana* corresponding to high values while other species corresponded to low

values (Supplementary Fig. 19c). Interestingly, GC content alone reproduced the distinct model behaviors while the separation between species was exacerbated for Genewiz (Supplementary Fig. 19d). Linear regressions for GC content of generated and natural sequences showed that CodonTransformer better fitted natural GC content (Supplementary Fig. 19e).

To conclude, sequences generated by CodonTransformer closely recapitulate natural distribution of low- and high-frequency codons as represented by %MinMax profiles and reproduce natural GC content and RNA folding energy.

## Model benchmark for heterologous expression of proteins

To further compare the performance of codon optimization tools in the context of heterologous expression, we collected 52 recombinant proteins with applications in molecular biology and therapeutics. Using each optimization tool, we designed DNA sequences for expression in *E. coli, S. cerevisiae, A. thaliana, M. musculus, and H. sapiens* ("Source Data File").

We first compared the sequences generated by different tools using a set-based similarity measure: the Jaccard index[23,43]. For each of the benchmark proteins, we calculated the Jaccard index *i.e.*, the similarity between two sequences as the ratio between the intersection and the union of the corresponding codon sets ("Methods"). A value close to 0 and 1 indicates a low and high similarity, respectively. We computed the mean value and standard deviation between analogous sequences for all tools in each organism (Supplementary Fig. 20) and across the organisms (Fig. 4a). Across organisms (excluding ICOR as it works only for *E. coli*), the base model showed a greater Jaccard similarity with the fine-tuned model (mean of 0.74) than with any other tools (Twist: 0.61, IDT: 0.60, Genewiz: 0.59). Among all the pairs, IDT and Twist showed the highest similarity score (0.92).

We then used a similarity measure that considers codon position: the sequence similarity. We calculated the sequence similarity as the percentage of matching codons ("Methods") and computed the mean value and standard deviation for all tools at the organism level (Supplementary Fig. 21) and across all organisms (Fig. 4b). CodonTransformer fine-tuned and base models showed the highest score when compared between them (mean of 69.26) and comparable values when compared to other models (maximal difference in mean <2). Compared to other models, IDT showed the overall lowest similarity score (mean <40 and standard deviation <4 for all comparisons) followed by Twist (mean <50 and standard deviation <4 for all comparisons). Genewiz showed relatively high similarity with both base and fine-tuned models (mean of 61.42 and 62.58 respectively), also with ICOR for which mean and standard deviation are computed only for *E. coli* genes.

To explain the differences between these two similarity measures, we looked at the codon distributions across the 52 sequences (Supplementary Fig. 22). For the 5 organisms, IDT and Twist generated sequences with more uniform codon distributions resulting in codons being present in a large number of sequences and explains the high Jaccard index between these models. The low sequence similarity suggests that the codons are positioned differently within the sequences. On the contrary, CodonTransformer and Genewiz displayed high differences in their total count of codons and number of sequences where a codon is present. These distinct model behaviors were also observed for the 50 natural sequences selected for each organism (Supplementary Fig. 23), which recapitulated distributions observed at the genome level (Supplementary Fig. 24).

Finally, we computed the %MinMax profiles and the normalized DTW distances for sequences generated by the different tools. In line with the results obtained for natural genes, (excluding ICOR as it works only for *E. coli*) we observed minimal DTW distance between the CodonTransformer base and fine-tuned model (mean of 0.05 ± 0.03) when computing the mean across organisms (Fig. 4c, Supplementary

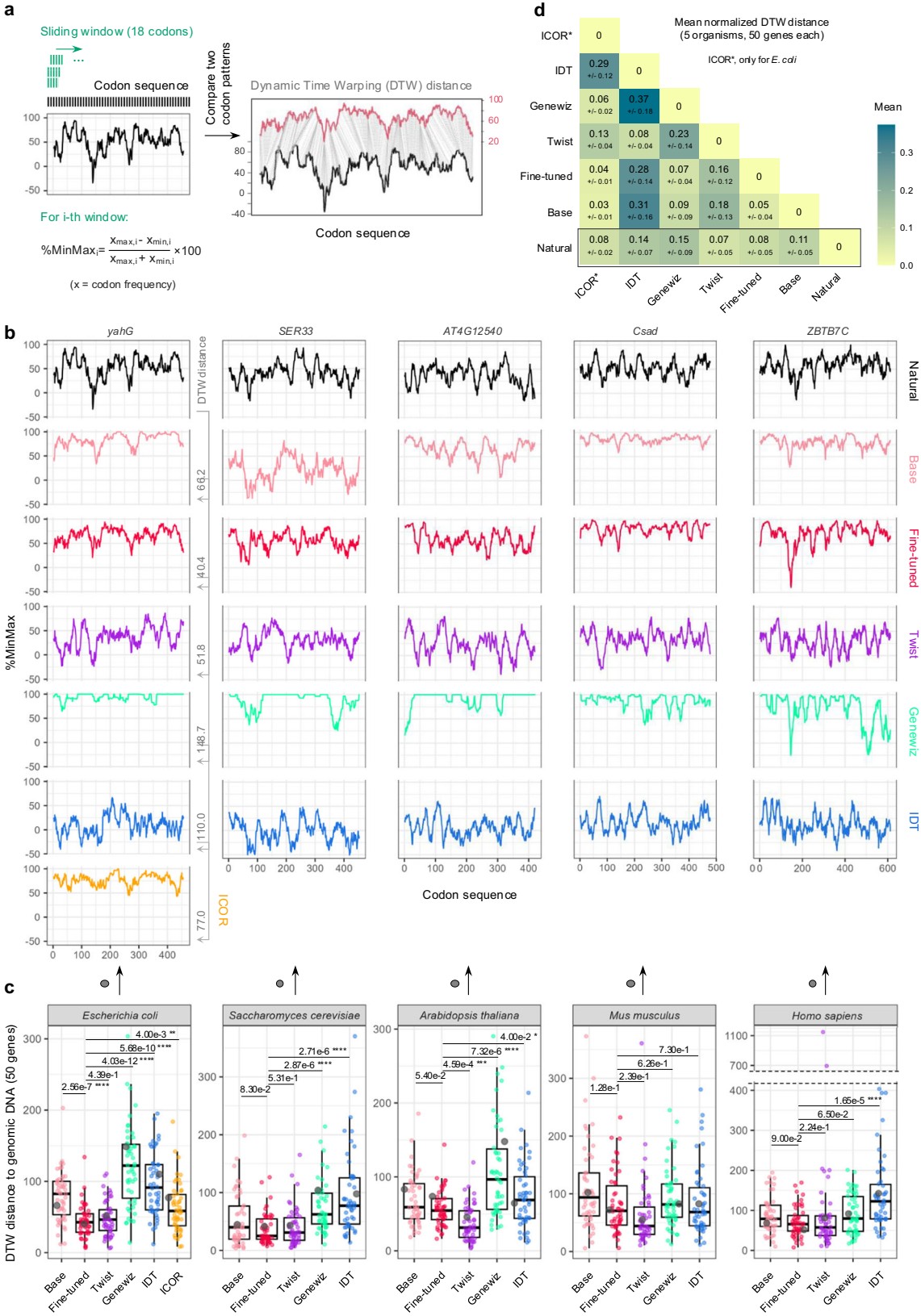

Fig. 25). Low DTW distance were also observed between Twist and IDT (mean of 0.07 ± 0.02) while Genewiz and Twist had intermediate distance (mean of 0.24 ± 0.11) and the highest DTW distance was between Genewiz and IDT (mean of 0.40 ± 0.15). Notably, the trend among the different tools for DTW distances of benchmark proteins (Fig. 4c) resembles the one observed for natural genes (Fig. 3d). Additionally,

observations made on RNA folding energy for natural proteins also hold true for the benchmark sequences (Supplementary Fig. 19). These suggest that CodonTransformer robustly designs sequences with natural-like codon distribution and RNA folding energy for new amino acid sequences beyond its training set making it a suitable tool for heterologous expression.

**Fig. 3 | CodonTransformer generates natural-like codon distributions.**
**a** Schematic representation of %MinMax and dynamic time warping (DTW). %
Minmax represents the proportion of common and rare codons in a sliding window
of 18 codons. DTW algorithm computes the minimal distance between two %Min-
Max profiles by finding the matching positions ("Methods"). **b** %MinMax profiles for
sequences generated by different models for genes *yahG* (*E. coli*), *SER33* (*S. cere-
visiae*), *AT4G12540* (*A. thaliana*), *Csad* (*M. musculus*), *ZBTB7C* (*H. sapiens*). **c** DTW
distances between %MinMax profiles of model-generated sequences and their
genomic counterparts for 50 random genes selected among the top 10% codon

similarity index (CSI). For each organism, the gene for which the %MinMax profiles
are represented above (**b**) is highlighted in gray. Mean DTW distances were com-
pared to the fine-tuned model using a two-sided unpaired t-test ($n = 52$), with the
numerical p-value shown for each. Center line shows the median; box limits
represent the 25th (Q1) and 75th (Q3) percentiles; whiskers extend to 1.5× inter-
quartile range (IQR); points are outliers beyond whiskers. **d** Mean and standard
deviation of normalized DTW distances by sequence length between sequences for
the 5 organisms (for organism-specific DTW distances, see Supplementary Figs. 18).
Data underlying this figure is provided in the Source Data File.

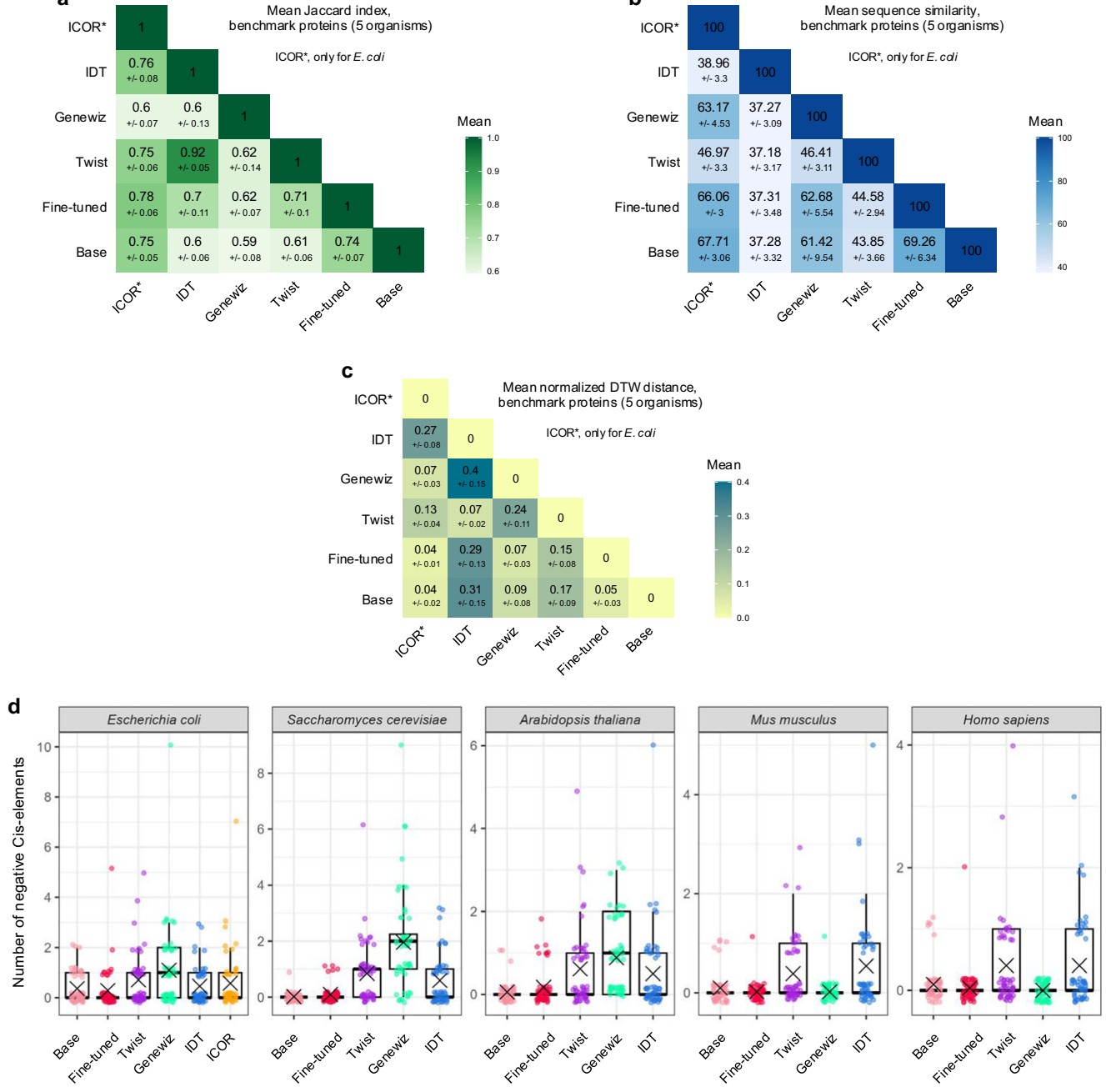

**Fig. 4 | Model benchmark with proteins of biotechnological interest.** Mean and
standard deviation of Jaccard index (**a**), sequence similarity (**b**), and dynamic time
warping (DWT) distance (**c**) between corresponding sequences for the 52 bench-
mark proteins across the 5 organisms (for organism-specific results, see Supple-
mentary Figs. 20, 21, and 25, respectively). **d** Number of negative cis-elements in the

52 sequences generated by different tools for each organism (X shows the mean).
Center line shows the median; box limits represent the 25th (Q1) and 75th (Q3)
percentiles; whiskers extend to 1.5x IQR; points are outliers beyond whiskers. Data
underlying this figure is provided in the Source Data File.

Altogether, these results suggest that heterologous sequences generated by codon optimization tools differ in their global and local codon usage and corroborate observations made on natural sequences.

## CodonTransformer generates sequences with minimal negative cis-elements

To minimize regulatory interference of the host organism on the expression of a heterologous gene of interest, negative cis-regulatory elements (*e.g.*, operators and silencers)[44–46] should be avoided in DNA sequence design[18,47]. We used a freely available tool developed by Genescript (https://www.genscript.com/tools/rare-codon-analysis) (as it has previously been used to analyze codon-optimized sequences[18,47]) to quantify the number of negative cis-elements in sequences designed by the different tools for the 52 benchmark proteins (Fig. 4d).

Among all models, Genewiz designed sequences with the highest number of negative cis-elements for some species (mean of 1.1 for *E. coli*, 1.96 for *S. cerevisiae* and 0.88 for *A. thaliana*), whereas the sequences contained almost zero for *M. musculus* and *H. sapiens*. Twist and IDT showed equivalent results with an intermediate number of cis-elements across organisms. Finally CodonTransformer, both base and fine-tuned models, robustly designed sequences with minimal negative cis-elements for all 5 organisms (mean of 0.37 and 0.29 for *E. coli*, 0.02 and 0.1 for *S. cerevisiae*, 0.06 and 0.17 for *A. thaliana*, 0.1 and 0.02 for *M. musculus*, and 0.1 and 0.04 for *H. sapiens*). Notably, CodonTransformer was not specifically trained to avoid negative cis-elements, we observed a similar trend for ICOR *i.e.*, a neural network-based model[18]. For *E. coli*-optimized benchmark proteins, the fine-tuned model reduced the number of negative cis-elements compared to the base model (Fig. 4d). Indeed, we found that *E. coli* genes with high CSI harbor substantially lower number of negative cis-elements than genes with low CSI (Supplementary Fig. 26). The *E. coli* genes with high number of negative cis-elements (low CSI genes), were markedly reduced in these elements when codon-optimized by CodonTransformer (Supplementary Fig. 26), suggesting that our models can also be used for improved over-expression of endogenous genes.

These results show that CodonTransformer, both base and fine-tuned models, robustly generate sequences with minimal negative cis-elements across organisms.

## Discussion

In this work, we describe a multispecies context-aware deep learning model, CodonTransformer, with the following advances: First, its multispecies training enables it to learn and generate codon-optimized sequences for a wide range of host organisms, including popular model organisms (Fig. 2a, Supplementary Figs. 2–16). Multispecies training also enabled us to increase the size of the training data compared to other approaches[18,24] and to learn universal underlying rules. Second, we showcased the fine-tuning step on the top 10% CSI natural genes showing that the model performances can be further modulated according to a user-defined gene set. We then showed that base and fine-tuned models can predict the effect of synonymous mutations, providing an additional feature in relevant applications (Fig. 2b). Third, the model's ability to learn long-range codon patterns results in DNA sequences with natural-like codon distributions (Fig. 3), avoids the potential pitfalls of clustered low-/high-frequency codons[18,19]. Fourth, CodonTransformer generates sequences with minimal number of negative cis-elements across organisms, minimizing potential interference with gene expression in the host organism (Fig. 4d).

While available tools support a limited number of genes and have different behaviors in codon usage, CodonTransformer serves as an open-source tool with a known mechanism (*i.e.*, deep neural networks) generating sequences with natural-like distribution of low- and high-frequency codons. It has been reported that such distribution is important for the dynamic of the translation, for the folding of secondary and tertiary structures, and for protein multimerization and assembly[48]. In contrast, clusters of slow and fast codons can lead to protein aggregation, misfolding, and degradation[48] or to an expression level reduced by post-transcriptional mechanisms[49]. Therefore, sequences generated by CodonTransformer are likely to optimally preserve protein structure and function while expression level can be controlled at the level of transcription (via promoters) and translation initiation (via the Shine Dalgarno and Kozak sequences in prokaryotes and eukaryotes, respectively)[4].

Our results demonstrate that the fine-tuning process further allows modulating the model outcomes (Fig. 2). Using the top 10% natural CSI genes generated sequences with more natural-like patterns (Fig. 3) and decreased the number of negative cis-elements in *E. coli* (Fig. 4d).

Our results demonstrate that the fine-tuning process further allows modulating the model outcomes (Fig. 2), tuning the distribution of codons (Supplementary Fig. 24), generating sequences with more natural-like patterns (Fig. 3) and reducing the number of negative cis-elements in *E. coli* (Fig. 4d). Additionally, CodonTransformer can be customized by the community for broader or more specialized tasks, *e.g.*, custom fine-tuning on a given gene set with desired property (specific metric/expression level) or on a specific protein family (*e.g.*, for de novo protein design[11–15]). This is of particular interest in addressing a pronounced bottleneck in the expressibility of de novo-designed proteins[14,16]. We therefore provide as open-access, the base and fine-tuned models, as well as a Python package for easy implementation and a Google Colab notebook for online usage through a user-friendly interface ("Code availability").

Our approach leverages valuable genomic data that have already been optimized by evolution across the tree of life. CodonTransformer was trained on 164 genomes not only increasing the size of the training data but also allowing it to learn both universal and species-specific rules and constraints underlying gene expression. The sequence encoding strategy and model architecture described here consider both amino acids and nucleotides while capturing positional dependencies and can be adapted for protein design[50] or to address different bottlenecks of biotherapeutics such as alternative splicing, miRNA targeting and immunogenicity[51]. Future studies can extend the size and diversity of training sequences to also consider regulatory elements involved in transcription and translation.

## Methods

### Data

A total of 1,001,197 DNA sequences were collected from NCBI resources from 164 organisms including Humans (*Homo sapiens*), thale cress (*A. thaliana*), *C. elegans*, *C. reinhardtii* and its chloroplast, Zebrafish (*D. rerio*), fruit fly (*D. melanogaster*), house mouse (*M. musculus*), tobacco (*N. tabacum*) and its chloroplast, *P. putida*, and baker's yeast (*S. cerevisiae*) from Eukaryotes, along with all species of the Entrobactreacea order such as *Escherichia coli* and selected species from Archea such as *T. barophilus* and *Sulfolobus solfataricus*. Depending on the organism, these DNA sequences came in the gene or CDS format and were translated to protein sequences using NCBI Codon Tables[52]. During preprocessing, only the DNA sequences with a length divisible by three, starting with a start codon, ending with a stop codon, and only having a single stop codon were chosen. We made this dataset available at https://zenodo.org/records/12509224.

### Model input

The input for our Transformer model is a tokenized sequence created using both the DNA and protein sequences. Representing the DNA as a sequence of codons and the protein as a sequence of amino acids allows us to define tokens that integrate both the codon and the amino acid. For example, a hypothetical protein sequence of "M A L W _", where "_" represents the end of the sequence, and the corresponding DNA sequence of "ATG GCC CTG TGG TAA" are tokenized as a sequence of 5 tokens: "[M_ATG] [A_GCC] [L_CTG] [W_TGG] [_TAA]".

During Masked Language Modeling (MLM), a given token with a known codon and amino acid is changed to an alternative token with a known amino acid but unknown codon. For example, masking [A_GCC] will yield the token [A_UNK]. During inference, since we only know the protein sequence, all input tokens will be of the tokens with the form [aminoacid_UNK]. In both scenarios, the model's objective is to predict the correct [aminoacid_codon] token for a given [amino acid_UNK] token.

This approach introduces a training scheme we call STREAM (Shared Token Representation and Encoding with Aligned Multimasking). Unlike traditional MLM methods that employ a single generic mask token, STREAM leverages the inherent alignment between DNA and protein sequences to create multiple, specialized mask tokens. Using tokens of the form [aminoacid_UNK], we maintain partial information during masking, allowing the model to simultaneously learn a shared representation of both DNA and protein sequences. This multi-masking strategy, combined with the natural alignment of the data, enables a more nuanced and context-aware learning process.

As a result, our vocabulary includes 64 tokens with both a known codon and a known amino acid, 21 tokens with an unknown codon and a known amino acid, and 5 special tokens including the general unknown token ([UNK]), the start of sequence token ([CLS]), the end of sequence token ([SEP]), padding token ([PAD]), and a general masking token ([MASK]). This will bring our total vocabulary size to 90 tokens.

The final input to the model is a truncated and padded sequence with a maximum length of 2048 tokens, that starts with a [CLS] token, includes the sequence tokens, ends with a [SEP] token, and is followed by several [PAD] tokens. A simple example could be "[CLS] [M_ATG] [A_GCC] [L_CTG] [W_TGG] [_TAA] [SEP] [PAD] ... [PAD]". In addition to the input sequence, the model also receives a taxonomy ID as input, a unique number defining the target organism. Subsequently, this organism ID will be considered as the token type ID of each token in the sequence for that organism.

## Embedding

The model learns an embedding vector for each token in the vocabulary, as well as for each organism ID, which is provided as a token type ID. During training, these token type embeddings are added to the learned embeddings for each token in the sequence[53] to integrate organism-specific information into the token representations. Additionally, the model incorporates positional embeddings that represent the absolute position of each token in the sequence, enabling the model to capture and leverage sequential dependencies within the input data.

## Model structure

Our model architecture is based on BigBird[29], a variant of the BERT Transformer[30], with a block sparse attention mechanism, enabling it to efficiently process much longer sequences than a standard BERT Transformer. Hence, CodonTransformer can optimize the DNA sequence for a protein sequence with a maximum length of 2048 tokens. CodonTransformer has 12 hidden layers, 12 attention heads, a hidden size of 768, an intermediate size of 3072, and an attention type of block sparse with a block size of 64, bringing the total number of parameters to 89.6 million. The model is curated using the open-source Transformers package from Hugging Face[54].

## Training

There are two stages in model training: pretraining and fine-tuning. The goal of the pretraining is to teach the model what a general input sequence looks like, while the fine-tuning focuses on adapting the model to specifically predict DNA sequences that are highly optimized for the target organism.

Both stages share the same MLM objective, in which 15% of tokens are randomly selected, and out of them, 80% are masked (*e.g.*, the chosen amino acid token with a known codon like [M_ATG] is swapped

with the corresponding amino acid token with an unknown codon, [M_UNK]), 10% are swapped with a random token, and 10% remain unchanged. Following this, the model has to predict the correct token that was masked.

Pretraining uses all sequences in the dataset. It uses a batch size of 6 and 16 NVIDIA V100 GPUs, for 5 epochs. The learning rate starts from 5e-7, linearly increases to 5e-5 over the first 10% of training steps, and then linearly decreases to 5e-7 over the remaining.

Fine-tuning follows a similar process to pretraining but uses a subset of the dataset. To ensure high optimization, we select the top 10% of genes with the highest codon similarity index (CSI) from various organisms: *E. coli, B. subtilis, P. putida, T. barophilus, S. cerevisiae, C. reinhardtii* (and its chloroplast), *N. tabacum* (and its chloroplast), *A. thaliana, C. elegans, D. rerio, D. melanogaster, M. musculus* and *H. sapiens*. The batch size and learning rate are the same as pretraining, and we use 4 NVIDIA V100 GPUs for 15 epochs. This fine-tuning step allows the model to learn the codon distribution patterns of highly optimized genes.

## Inference

Inference was conducted using two primary methods:

Protein sequence only: In this method, the model is given only the protein sequence. Each amino acid is represented by tokens in the format [aminoacid_UNK]. The model then processes these tokens to unmask and convert them into the [aminoacid_codon] format, thereby generating the corresponding DNA sequence.

Protein with DNA sequence: In this method, the model is provided with both the protein sequence and a plausible DNA sequence. Since both are provided, the sequence is initially encoded using [aminoacid_codon] tokens. The model optimizes these tokens by replacing them with more suitable [aminoacid_codon] tokens for the given protein sequence.

Following, the codon parts from all tokens are concatenated to produce the predicted DNA sequence.

## Custom fine-tuning

Fine-tuning customizes the base CodonTransformer model to a specific set of genes. This process, exemplified by our use of sequences with high (top 10%) codon similarity index (CSI), enhances the model's performance for optimizing DNA sequences for various species and conditions. The CodonTransformer repository provides a guide on fine-tuning the base model ("Code availability").

## Evaluation metrics

Several metrics are used to assess codon-optimized sequences, providing a comprehensive view of codon usage and sequence properties:

## Codon similarity index (CSI)

CSI[25] is an application of CAI[8] to the codon usage table of an organism (representing the codon frequency across the entire genome). It has the advantage that it does not rely on the arbitrary selection of a reference gene set. The rationale is that codon optimization based on the most frequent codons (as determined from highly expressed genes) may be detrimental to the host and impair the correct protein folding. Therefore, CSI may provide a softer estimator of codon usage for a specific organism. For the computational implementation of the CSI calculations, we utilized the CAI and CAI-PyPI Python libraries[55], for which we used the overall codon usage tables[56].

The relative adaptiveness of a codon $w_{ij}$ is calculated as the ratio of its frequency $x_{ij}$ to one of the most used codon $x_{imax}$ for the same amino acid.

$$w_{ij} = \frac{x_{ij}}{x_{imax}} \tag{1}$$

The CSI for a gene is the geometric mean of these relative adaptiveness values across all codons in a DNA sequence (gene) with a length of $L$ codons:

$$\text{CSI} = \exp\left(\frac{1}{L}\sum_{k=1}^{L} \ln w_k\right) \tag{2}$$

### Codon frequency distribution (CFD)

The CFD quantifies the percentage of rare codons (those used less than 30% as often as the most frequent codon) in a gene. A weight $w_{ij}$ is assigned to each codon, where $w_{ij} = 1$ if the codon is rare and 0 otherwise:

$$w_{ij} = \left\{ 1 \text{ if } \frac{x_{ij}}{x_{imax}} < 0.30, \ 0 \text{ otherwise} \right\} \tag{3}$$

The CFD is the mean of these weights:

$$\text{CDF} = \frac{1}{L}\sum_{k=1}^{L} w_k \tag{4}$$

### GC content (%GC)

GC content measures the proportion of guanine (G) and cytosine (C) bases in a DNA sequence:

$$\text{GC} = \frac{G + C}{A + T + G + C} \tag{5}$$

### Negative cis-regulatory elements

The number of negative cis-regulatory elements was determined using the Genscript codon analysis tool (https://www.genscript.com/tools/rare-codon-analysis).

### %MinMax

The %MinMax metric[41] evaluates the balance between high and low frequency codons within a window of specific length sliding along the sequence.

For a given window size of $w$ ($w = 18$ as previously reported[41]), the %MinMax for each window $i$ is calculated as:

$$\%\text{MinMax}_i = \frac{x_{\max,i} - x_{\min,i}}{x_{\max,i} + x_{\min,i}} \times 100 \tag{6}$$

where $x_{\max,i}$ and $x_{\min,i}$ are the maximum and minimum codon usage frequencies within the $i$-th window, respectively. The overall %MinMax for the gene is represented as an array of %MinMax values for each position of the window:

$$\%\text{MinMax} = \left[\%\text{MinMax}_1, \%\text{MinMax}_2, \ldots, \%\text{MinMax}_n\right] \tag{7}$$

where $n$ is the number of windows in the sequence.

To compute minmax profiles, we used the R package kodonz available at https://github.com/HVoltBb/kodonz/ using built-in codon usage tables for the different organisms.

### Dynamic time warping (DTW)

DTW is an algorithm for measuring the similarity between two temporal sequences or shape matching[57,58]. We employ DTW to quantify distances in %MinMax of codon usage patterns. DTW calculates the optimal alignment between two time series $X$ and $Y$, as:

$$\text{DTW}(X, Y) = \sqrt{\sum_{(i,j)\in\pi}|\boldsymbol{X}_i - \boldsymbol{Y}_j|^2} \tag{8}$$

Where $X$ and $Y$ are the %MinMax arrays, $\pi$ is the optimal alignment path, $X_i$ and $Y_j$ are aligned points in the sequences, and $\|X_i - Y_j\|$ is the Euclidean distance between these points. This metric enables quantitative assessment of how closely one sequence's codon usage pattern aligns with that of a target sequence.

To compute the DTW distances, we used the main function from the R package dtw[59] with default parameters and retrieved from the result object the distance or normalizedDistance items.

### Jaccard index

We used the Jaccard index to evaluate the propensity of models to use the same codons in the generated sequences. Jaccard index is a set-based similarity measure that evaluates the similarity between two sets as the ratio between their intersection and their union.

For two DNA sequences $X$ and $Y$ composed of codon sets A and B, the Jaccard index is as follows:

$$J(X, Y) = \frac{A \cap B}{A \cup B} \tag{9}$$

### Sequence similarity analysis

Sequence similarity measures the percentage of codons matching two sequences. This metric is crucial for assessing the functional and evolutionary relationships between genes.

For sequences $A$ and $B$, the sequence similarity is calculated as:

$$\text{Similarity}(A, B) = \frac{1}{L}\sum_{i=1}^{L} \delta(a_i, b_i) \times 100 \tag{10}$$

where $L$ is the length of the sequences, $a_i$ and $b_i$ are the codons at position $i$ in sequences $A$ and $B$, respectively, and $\delta$ is the Kronecker delta function, which is 1 if $a_i = b_i$ and 0 if $a_i \neq b_i$. This metric ranges from 0% to 100%, with 100% indicating identical sequences and 0% indicating completely different sequences.

### Minimum free energy of RNA structures

To compute the minimum free energy for the sequences generated by the models, we translated those sequences to RNA using the R package XNAString[60] with default sugar and backbone structures. This package then calls the RNAfold_MFE function from ViennaRNA package[42] to compute the minimal folding energy for the RNA structure. The R code to calculate %MinMax, DTW, Jaccard index, sequence similarity and RNA minimum free energy has been provided with the Source Data.

### Reporting summary

Further information on research design is available in the Nature Portfolio Reporting Summary linked to this article.

## Data availability

All genomic data (164 organisms) used to train the model are available at both https://zenodo.org/records/12509224 and https://huggingface.co/datasets/adibvafa/CodonTransformer. Data for Fig. 2a and Supplementary Figs. 2–16 are available at https://zenodo.org/records/13262517. Data for Figs. 2b, 3, 4, Supplementary Figs. 18–25, and the custom code used to produce results in this work are available as Source Data File. Source data are provided with this paper.

## Code availability

The user-friendly Google Colab Notebook for codon optimization can be accessed here. The CodonTransformer model and codes used to

train and evaluate models are available at the CodonTransformer GitHub repository: https://github.com/Adibvafa/CodonTransformer[61] under Apache-2.0 License. Base and fine-tuned model weights are available at Hugging Face, https://huggingface.co/adibvafa/CodonTransformer, along with a guide that explains how CodonTransformer package can be used to fine-tune the base model on a custom dataset. CodonTransformer package: We introduced a comprehensive Python Package named CodonTransformer that supports the development and evaluation of such models. The toolkit available at https://pypi.org/project/CodonTransformer/ includes 5 major modules: CodonData: Simplifies the process of gathering, preprocessing, and representing data to achieve a clean and well-structured dataset. CodonPrediction: Enables easy tokenization, loading, and utilization of CodonTransformer for making predictions. It also includes various other approaches such as High-Frequency Choice (HFC), Background Frequency Choice (BFC), Uniform Random Choice (URC), and ICOR. CodonEvaluation: Offers scripts to evaluate various metrics for codon-optimized sequences. CodonUtils: Provides essential utility functions and constants that streamline the workflow. CodonJupyter: Comes with tools for creating demo notebooks, allowing users to quickly set up and demonstrate the capabilities of CodonTransformer in an interactive environment.

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

## Acknowledgements

This work was supported by the University of Toronto Excellence Award (UTEA) provided by the University of Toronto Scarborough (A.F.), and NSERC (Discovery Grant RGPIN-2020-06377), CIHR (Project Grant FRN-192109) and the University of Toronto (G.J.F.), the MOPGA (Make Our Planet Great Again) Young Researcher Fellowship by the French Government and ATIP-Avenir Research Group Leader Program by Inserm-CNRS (A.P.), the Fondation Bettencourt Schueller (A.P. and A.B.L.). The authors wish to thank Ali Yazdizadeh Kharrazi, Amir Zare, Aude Bernheim, Ernest Mordret, Mike Schäkermann, Rajeev Mylapalli, and Vincent Libis for their insights, feedback and fruitful discussions.

## Author contributions

A.P. conceived the study. A.F. constructed the model with the design and under the supervision of G.J.F. A.F. and V.G. performed the simulations. A.P., V.G., A.F., A.B.L., and G.J.F. analyzed the results. A.P., V.G., and A.F. drafted the manuscript with support from G.J.F. and A.B.L. All authors approved the final draft.

## Competing interests

The authors declare no competing interests.
