## [Transparent Peer Review file · Nature Communications]

CodonTransformer: a multispecies codon optimizer using context-aware neural networks

Corresponding Author: Dr Amir Pandi

Version 0:

Reviewer comments:

Reviewer #1

(Remarks to the Author)

The article by Fallahpour et al presents CodonTransformer, an masked language modelling approach for codon optimization that has been trained on 1 million sequences from 164 common organisms. The model uses a clever tokenization scheme where every amino acid can be associated with a cognate codon, or to a mask token, and the prediction of the codon token enacts a bidirectional translation process. The authors test the model on a variety of synthetic benchmarks, and show among others that the models naturally learn to introduce codons with the right translations speeds (as measured by %MinMax), and that provide optimal RNA folding free energies.

In my opinion, the paper is beautifully written and technically sound, and it introduces an elegant and refreshing way of studying a very important problem that is central to biology -- if sometimes somewhat ignored. One of the drawbacks of the paper is that it does not provide an experimental validation of whether heterologous expression benefits from this codon optimization approach -- but I think it would be unfair to judge the merit of the paper on the author's experimental capabilities. In my opinion, this article would be a good fit for Nature Communications. I would like to recommend this work for publication, subject to the authors addressing a few questions below.

- I am not sure that I find the CSI benchmark particularly meaningful -- unless I misunderstood the definition, a trivial model that randomly samples from the codon frequency table of an organism would achieve a CSI near 1, whereas we must agree such model would not be very helpful. Even a 3-mer prediction model (predict the most common codon for an amino acid, based on the codons that surround it) should achieve a pretty good score on this benchmark. Could the authors perform a comparison against these two extremely simple cases, and explain why the CSI is significant?

- I am not sure if I understand the purpose of the first part of the section "Model benchmark with proteins of biotechnological interest". The comparison of the %MinMax and the RNA folding free energy are reasonable comparisons, as they relate to well-known factors in protein expression. Personally, I find their results really enticing. However, the algorithms the authors are comparing against are not necessarily "optimal", and in fact many of them will include extraneous considerations such as eliminating repeated sequences that hinder DNA synthesis. Could the authors provide a better explanation of why comparing with Twist's and Genewiz's algorithms is meaningful? Moreover, is there any way that the authors could explain *how* their algorithm is different, even if it just through a couple of case studies?

- The ultimate question in codon optimization is not whether we can recover the sequence of an organism (even though it is a good benchmark!), but whether we can perform heterologous expression successfully. Unfortunately, testing this definitively requires experimental validation, which is costly, time-consuming and not widely accessible to computational researchers. I am wondering, however, if the authors could use a simple benchmark to validate whether the model has any useful understanding of the factors behind protein expression yield. For example, they could take a paper that has compared multiple synonymous constructs expressing the same protein (e.g. some of the papers from Patricia Clark's group on codon harmonization), and observe if there is any meaningful correlation between the log-likelihood of the sequence predicted by CodonTransformer, and the expression levels. Could the authors provide some sort of benchmark like this?

I look forward to reading the authors' responses to my questions and look forward to reaching a final decision of acceptance. In any case, I would like to commend the authors on a very interesting paper and an excellent codebase.

Carlos Outeiral

(Remarks on code availability)

I would like to commend the authors for the quality of their codebase and the accompanying documentation.

Reviewer #2

(Remarks to the Author)

Fallahpour et al. present a context-aware neural network-based multispecies codon optimizer. While the manuscript is overall well-written, there're still several issues to be addressed before moving on:

1. How and why did the authors choose the 164 organisms for training CodonTransformer? Is there any possibility for species-bias (e.g. due to the imbalance gene counts across species)?
2. In Figure 2, the pattern of genomic DNA and generated DNA sequences has obvious difference. How these results could be used to support the point "...efficiently learned codon usage preferences across organisms..." (line 150-151)?
3. In Figure 3, why only select 50 random genes? How did other methods generate sequences? In particular, how did the author select these 5 genes for demonstration in Figure 3b? And statistical testing results should be presented in Figure 3c.
4. In section "Model benchmark with proteins of biotechnological interest", the author compared sequences generated by different tools with each other rather than "ground truth", which may be fine per se but this referee would question that such result could be used to support author's claim "...CodonTransformer extrapolates natural-like features to new sequences" (line 245).
5. The last but not the least, this referee would suggest a real-world case for demonstrating the superiority (and usage) of CodonTransformer.

(Remarks on code availability)

The GitHub repo is overall well-organized and actively maintained.

Reviewer #3

(Remarks to the Author)

The relevance of this subject is limited for a broad audience, but it is still an interesting and technically valuable contribution.

This paper presents CodonTransformer, a novel approach to codon optimization using a deep learning model based on Transformer architecture. The study results appear comparable to or better than previous approaches.

I appreciate the generalization of codon optimization across 164 organisms, covering a wide variety of species from different kingdoms of life, including both eukaryotes and prokaryotes. This diversity in training data is commendable and represents a substantial leap over single-species optimization approaches.

However, some aspects still need to be clarified. For example, does CodonTransformer focus primarily on subsets of codons with higher frequency, as the violin plots suggest (if I am not mistaken)? There appears to be a discrepancy between the spread of genomic data and the generated sequences—could this indicate a bias in the model toward certain codon choices? Clarifying this would strengthen the paper's claims about producing natural-like codon distributions.

Additionally, it would be helpful to know how organism-specificity is encoded. Is the organism specification introduced via one-hot encoding, or is there a more complex embedding scheme in use? Understanding this would provide better insight into how the model handles multispecies optimization.

Another important aspect to explore is the intra- and inter-species variability in codon usage. Does CodonTransformer accurately capture this variability, or does it tend to amplify or smooth out these differences? This point deserves further elaboration and validation within the study.

(Remarks on code availability)

Version 1:

Reviewer comments:

Reviewer #2

(Remarks to the Author)

This referee would thank for the authors' extensive works which have effectively addressed previous concerns. Now this referee believes the manuscript is ready to move forward.

The comments below are based on personal understanding of Reviewer 1's comments and author's responses:

The authors have generally addressed almost all comments from Reviewer 1. Specifically, I still find the last comment from Reviewer 1 (-The ultimate question in codon optimization is not whether we can recover ...) was not well-explored. I understand the difficulty in wet-experiment, so I think this's acceptable so far.

(Remarks on code availability)

The GitHub repo is overall well-organized and actively maintained.

Reviewer #3

(Remarks to the Author)

All my issues have been addressed. I have no more points to raise.

(Remarks on code availability)

Reviewer #4

(Remarks to the Author)

(Remarks on code availability)

REVIEWER COMMENTS

We thank all the Reviewers for their thorough review of our work and constructive comments, which helped us improve our manuscript. We took all their comments into consideration as addressed below, highlighting as well the changes made throughout the manuscript.

Reviewer #1 (Remarks to the Author):

The article by Fallahpour et al presents CodonTransformer, an masked language modeling approach for codon optimization that has been trained on 1 million sequences from 164 common organisms. The model uses a clever tokenization scheme where every amino acid can be associated with a cognate codon, or to a mask token, and the prediction of the codon token enacts a bidirectional translation process. The authors test the model on a variety of synthetic benchmarks, and show among others that the models naturally learn to introduce codons with the right translations speeds (as measured by %MinMax), and that provide optimal RNA folding free energies.

In my opinion, the paper is beautifully written and technically sound, and it introduces an elegant and refreshing way of studying a very important problem that is central to biology -- if sometimes somewhat ignored. One of the drawbacks of the paper is that it does not provide an experimental validation of whether heterologous expression benefits from this codon optimization approach -- but I think it would be unfair to judge the merit of the paper on the author's experimental capabilities. In my opinion, this article would be a good fit for Nature Communications. I would like to recommend this work for publication, subject to the authors addressing a few questions below.

Response to Reviewer #1:

We thank the Reviewer for his positive feedback on our work and for raising key points regarding the expected CSI values for simple models, the comparison of commercial codon optimization algorithms and predicting synonymous mutations, which we addressed in the revised manuscript. Below, we provide point by point responses (in blue) and actions (in brown) with changes in the main text (underlined).

- I am not sure that I find the CSI benchmark particularly meaningful -- unless I misunderstood the definition, a trivial model that randomly samples from the codon frequency table of an organism would achieve a CSI near 1, whereas we must agree such model would not be very helpful. Even a 3-mer prediction model (predict the most common codon for an amino acid, based on the codons that surround it) should achieve a pretty good score on this benchmark. Could the authors perform a comparison against these two extremely simple cases, and explain why the CSI is significant?

We used the CSI as a similarity measure to the organismic codon usage table, providing a metric for comparison across organisms. A sequence composed only of the most frequent codons would indeed yield a CSI value of 1. Thus, the CSI does not measure the optimality of a sequence, for which codon pattern within each sequence should be analyzed, as we do in Fig. 3. In Fig. 2 and Supplementary Figs. 2-16, we show that the base model, trained on 164 genomes, generates for the 15 organisms sequences with a high CSI score, demonstrating that the base model learned codon preferences across organisms.

We reworked the results section (L115-146) to remove previous ambiguity and clarify the rationale for using the CSI. To illustrate the behavior of the CSI, we compared the values of sequences generated by CodonTransformer to two simple models. The uniform random choice

(URC) model picks random codons for each amino acid, while background frequency choice (BFC) selects codons according to their frequency in the organismic codon usage table (revised Supplementary Figs. 2-16). BFC generates sequences with a CSI that is similar to that of genomic sequences, while URC generates sequences with a lower CSI. CodonTransformer generated sequences with higher CSI than these two models.

Hence we used CSI as a similarity metric to demonstrate that the implemented organism encoding allows CodonTransformer to learn, as a single model, species-specific codon preferences (revised Fig. 1c).

[L115-146]:

We trained the base model, which we named CodonTransformer, using ~1 million genes from 164 organisms (**Data availability**). The training set is a collection of genomes from all domains of life, i.e., Bacteria, Archaea, and Eukarya that constitute 56.1%, 2.5% and 41.4% of sequences, respectively. This model can be either used directly for codon optimization across species or it can be fine-tuned on custom sets of DNA sequences to perform more tailored tasks. In this study, we fine-tuned CodonTransformer on the 10% genes with the highest CSI (Methods) for 15 organisms: *Escherichia coli*, *Bacillus subtilis*, *Pseudomonas putida*, *Thermococcus barophilus*, *Saccharomyces cerevisiae*, *Chlamydomonas reinhardtii* and its chloroplast, *Arabidopsis thaliana*, *Nicotiana tabacum* and its chloroplast, *Caenorhabditis elegans*, *Danio rerio*, *Drosophila melanogaster*, *Mus musculus*, and *Homo sapiens* (Fig. 1c). The CSI²⁵ is derived from the codon adaptation index (CAI)⁸ to quantify the similarity of codon usage between a sequence and the codon usage frequency table of an organism instead of an arbitrary reference set of highly expressed genes. It therefore provides a robust metric at the multispecies level and may constitute a superior predictor of expression level in higher eukaryotes^{4,7,25,31}.

CodonTransformer learned codon usage across organisms

To assess the ability of the model to capture organism-specific codon preferences, we generated DNA sequences for all proteins encoded by the 15 genomes for which we performed fine-tuning (Fig. 1c). We then compared the sequences generated by the base and fine-tuned models to their natural counterparts (entire genome and top 10% CSI used for fine-tuning).

Sequences generated by the base model show a higher percentage of matching codons with their natural DNA counterparts than randomly selected codons without and with organismic preferences, uniform random choice (URC) and background frequency choice (BFC), respectively (Supplementary Fig. 1). Those sequences have a high CSI indicating that they follow for each organism the preference of codon usage (**Fig. 2, Supplementary Figs. 2-16**). Obtaining higher CSI than genomic sequences is expected because not every DNA sequence of a species is tuned for optimal expression, whereas CodonTransformer optimizes every sequence based on codon frequencies that it encounters during training. The base model generated sequences with higher CSI than the top 10% genomic CSI for all organisms except *S. cerevisiae*, *N. tabacum* and its chloroplast (**Fig. 2, Supplementary Figs. 2-16**). When fine-tuning the model, CodonTransformer generated sequences with lower CSI than the base model for all organisms except *S. cerevisiae*, *N. tabacum* and its chloroplast which showed an increase in the CSI better mimicking the top 10% genomic CSI on which they were fine-tuned (Fig. 2, Supplementary Figs. 2-16). As expected, the clustering of CodonTransformer embeddings for organisms used for training resembles their overall phylogenetic distances (Supplementary Fig. 17).

- I am not sure if I understand the purpose of the first part of the section "Model benchmark with proteins of biotechnological interest". The comparison of the %MinMax and the RNA folding free energy are reasonable comparisons, as they relate to well-known factors in protein expression. Personally, I find their results really enticing. However, the algorithms the authors are comparing against are not necessarily "optimal", and in fact many of them will include extraneous considerations such as eliminating repeated sequences that hinder DNA synthesis. Could the authors provide a better explanation of why comparing with Twist's and Genewiz's algorithms is meaningful? Moreover, is there any way that the authors could explain *how* their algorithm is different, even if it just through a couple of case studies?

We modified the title of this section to "Model benchmark ~~with~~ for heterologous expression of proteins ~~of biotechnological interest~~" to emphasize the "heterologous" nature of those sequences (L237) and modified the introductory paragraph L238-241 to more clearly describe the aim of this section. We also added model comparison with respect to used codons for benchmarked and natural sequences revised Supplementary Figs. 22 and 23.

Codon optimization is often used for heterologous expression of proteins in a new host, which requires synthesis of new DNA. We therefore compared the tools offered by different DNA synthesis companies as they are broadly used in research. These tools are proprietary and little is known about the underlying algorithms. We first compared, for 5 organisms, 50 natural sequences to optimized sequences (previous result section). This provided insights into the behaviors of the models with %MinMax profiles suggesting that Genewiz maximizes the presence of frequent codons along the sequence, RNA free energy suggesting that IDT and Twist minimize folding energy for long sequences, and GC content differences being underestimated by IDT and exaggerated by Genewiz.

To be able to compare different models on common heterologous sequences, we selected 52 proteins with biotechnological applications and compared their optimized sequences using two similarity measures: Jaccard index which compares the "codon repertoire" and sequence similarity which is a position-wise comparison. A closer look at codon distributions for the 52 benchmark sequences in five organisms (revised Supplementary Figs. 22, see the *E. coli* example below) shows that Twist and IDT tend to pick more uniformly among codons leading most codons to be present in a high number of sequences (which explains the high Jaccard index but low sequence similarity between these two models, Fig. 4a and 4b). On the other hand, our model and Genewiz harbor a more discriminative selection among favorable codons (which may explain the low Jaccard index but high sequence similarity, Fig. 4a and 4b). Similar distributions were also observed for natural sequences (revised Supplementary Figs. 23, see the *E. coli* example below). We added a paragraph to discuss these results in the corresponding results section L261-269.

[L237-241]:

Model benchmark for heterologous expression of proteins

To further compare the performance of codon optimization tools in the context of heterologous expression, we collected 52 recombinant proteins with applications in molecular biology and therapeutics (**Supplementary Data 2**). Using each optimization tool, we designed DNA sequences for expression in *E. coli*, *S. cerevisiae*, *A. thaliana*, *M. musculus*, and *H. sapiens*. (**Supplementary**

Data 2).

[L261-269]:

To explain the differences between these two similarity measures, we looked at the codon distributions across the 52 sequences (**Supplementary Fig. 22**). For the 5 organisms, IDT and Twist generated sequences with more uniform codon distributions resulting in codons being present in a large number of sequences and explains the high Jaccard index between these models. The low sequence similarity suggests that the codons are positioned differently within the sequences. On the contrary, CodonTransformer and Genewiz displayed high differences in their total count of codons and number of sequences where a codon is present. These distinct model behaviors were also observed for the 50 natural sequences selected for each organism (**Supplementary Fig. 23**) which recapitulated distributions observed at the genome level (**Supplementary Fig. 24**).

- The ultimate question in codon optimization is not whether we can recover the sequence of an organism (even though it is a good benchmark!), but whether we can perform heterologous expression successfully. Unfortunately, testing this definitively requires experimental validation, which is costly, time-consuming and not widely accessible to computational researchers. I am wondering, however, if the authors could use a simple benchmark to validate whether the model has any useful understanding of the factors behind protein expression yield. For example, they could take a paper that has compared multiple synonymous constructs expressing the same protein (e.g. some of the papers from Patricia Clark's group on codon harmonization), and observe if there is any meaningful correlation between the log-likelihood of the sequence predicted by CodonTransformer, and the expression levels. Could the authors provide some sort of benchmark like this?

We thank the Reviewer for this comment and his guidance on how to predict synonymous mutations using CodonTransformer. We were not able to find synonymous mutation data from the codon harmonization work (doi.org/10.1002/pro.4223) suggested by the Reviewer. From the Clark group, we could only find such mutation data in a recent study that reported only mutations in the neighboring gene (doi.org/10.1073/pnas.2405510121) not suiting our aim here. In attempt to gather experimental data, we found a dataset of mutations in *ccdA*, *i.e.*, the antitoxin component of the *Escherichia coli* *ccdAB* toxin system, from a recent study (Chandra *et al.* 2022, *Molecular Biology and Evolution*, doi.org/10.1093/molbev/msac187). This data includes 62 synonymous mutations with their respective relative fitness and ribosome stalling values. We added a paragraph for the prediction results on this data and a new figure (2b) as follows:

[L167-177]:

Although we developed CodonTransformer for sequence design tasks, it can also be used to predict the effect of synonymous mutations. To do this, the probability of mutant and wild-type

codons can be predicted by the model for every amino acid position. We used 62 synonymous mutations in *ccdA*, i.e., the antitoxin component of the *E. coli ccdAB* toxin-antitoxin system, from a recent study³² with their respective experimental relative fitness and ribosome stalling values. Both base and fine-tune CodonTransformer models predicted the log-likelihood of the mutant codon significantly correlated with experimental relative fitness in a zero-shot manner with the wild-type DNA sequence as input to the model (Fig. 2b). The same prediction task by BFC that selects codons based on organismic codon usage table, also resulted in significant correlations slightly less than CodonTransformers. However, among BFC, base and fine-tuned CodonTransformer, only the fine-tuned model showed a significant correlation between log-likelihood of mutation and relative ribosome stalling (Fig. 2b).

[L320-322]:

We then showed that base and fine-tuned models can predict the effect of synonymous mutations, providing an additional feature in relevant applications (Fig. 2b).

I look forward to reading the authors' responses to my questions and look forward to reaching a final decision of acceptance. In any case, I would like to commend the authors on a very interesting paper and an excellent codebase.

We would like to once again thank the Reviewer for his positive feedback and constructive suggestions that helped improve our work. We hope that the Reviewer finds his points addressed via our responses and actions.

Reviewer #2 (Remarks to the Author):

Fallahpour et al. present a context-aware neural network-based multispecies codon optimizer. While the manuscript is overall well-written, there're still several issues to be addressed before moving on:

Response to Reviewer #2:

We thank the Reviewer for their comments on our approach and results including data balance in the model, statistical testing and a real-world show-case. Addressing these points further helped us improve our work during revision. Below, we provide point by point responses (in blue) and actions (in brown) with changes in the main text (underlined).

1. How and why did the authors choose the 164 organisms for training CodonTransformer? Is there any possibility for species-bias (e.g. due to the imbalance gene counts across species)?

Our data is a collection of genomes from all domains of life, Bacteria, Archaea, and Eukarya that constitute 56.1%, 2.5% and 41.4% of the genomic sequences, respectively. We added this information in the main text (L116-118) and Fig. 1c.

The species were subjectively chosen based on their significant use as model organisms (15 genomes that we used for fine-tuning, Fig. 2c) building up around half a million genes. Since the majority of model organisms were eukaryotes, which generally contain more coding sequences than prokaryotes, we added the same size of genomic data (around half a million) from bacterial and archaeal genomes to balance our training data. We used a novel organism encoding within our model, allowing each sequence to be customized for the associated organism, while the entire training set contributes to augmenting the training data as is essential for transformer-based models. This feature enables the model to learn and generate organism-specific sequences (Supplementary Fig. 2-16).

We took this comment as an opportunity to extend the discussion in L354-355.

[L116-118]:

The training set is a collection of genomes from all domains of life, i.e., Bacteria, Archaea, and Eukarya that constitute 56.1%, 2.5% and 41.4% of sequences, respectively.

[L354-355]:

Future studies can extend the size and diversity of training sequences to also consider regulatory elements involved in transcription and translation.

2. In Figure 2, the pattern of genomic DNA and generated DNA sequences has obvious difference. How these results could be used to support the point "...efficiently learned codon usage preferences across organisms..." (line 150-151)?

As mentioned in our response to the first point of Reviewers #1 and #3, high CSI values show that, for each organism, the codon usage in the sequences is close to the codon frequencies over the entire genome. When re-encoding natural proteins, the base model, after single multi-species training, generated novel DNA sequences with high CSI as codons with higher frequency were more encountered during training. The resulting high CSI across 15 genomes demonstrates that

the model efficiently learned the codon preferences of each organism (the aim of Fig. 2). Obtaining higher CSI than genomic sequences is expected because not every DNA sequence of a species needs to be optimally expressed, whereas CodonTransformer optimizes every sequence based on codon frequencies that it encounters during training. We modified the relevant sections in the main text (L115-146) to clarify this point.

[L115-146]:

We trained the base model, which we named CodonTransformer, using ~1 million genes from 164 organisms (**Data availability**). The training set is a collection of genomes from all domains of life, i.e., Bacteria, Archaea, and Eukarya that constitute 56.1%, 2.5% and 41.4% of sequences, respectively. This model can be either used directly for codon optimization across species or it can be fine-tuned on custom sets of DNA sequences to perform more tailored tasks. In this study, we fine-tuned CodonTransformer on the 10% genes with the highest CSI (Methods) for 15 organisms: *Escherichia coli*, *Bacillus subtilis*, *Pseudomonas putida*, *Thermococcus barophilus*, *Saccharomyces cerevisiae*, *Chlamydomonas reinhardtii* and its chloroplast, *Arabidopsis thaliana*, *Nicotiana tabacum* and its chloroplast, *Caenorhabditis elegans*, *Danio rerio*, *Drosophila melanogaster*, *Mus musculus*, and *Homo sapiens* (Fig. 1c). The CSI²⁵ is derived from the codon adaptation index (CAI)⁸ to quantify the similarity of codon usage between a sequence and the codon usage frequency table of an organism instead of an arbitrary reference set of highly expressed genes. It therefore provides a robust metric at the multispecies level and may constitute a superior predictor of expression level in higher eukaryotes^{4,7,25,31}.

CodonTransformer learned codon usage across organisms

To assess the ability of the model to capture organism-specific codon preferences, we generated DNA sequences for all proteins encoded by the 15 genomes for which we performed fine-tuning (Fig. 1c). We then compared the sequences generated by the base and fine-tuned models to their natural counterparts (entire genome and top 10% CSI used for fine-tuning).

Sequences generated by the base model show a higher percentage of matching codons with their natural DNA counterparts than randomly selected codons without and with organismic preferences, uniform random choice (URC) and background frequency choice (BFC), respectively (Supplementary Fig. 1). Those sequences have a high CSI indicating that they follow for each organism the preference of codon usage (Fig. 2, Supplementary Figs. 2-16). Obtaining higher CSI than genomic sequences is expected because not every DNA sequence of a species is tuned for optimal expression, whereas CodonTransformer optimizes every sequence based on codon frequencies that it encounters during training. The base model generated sequences with higher CSI than the top 10% genomic CSI for all organisms except *S. cerevisiae*, *N. tabacum* and its chloroplast (**Fig. 2, Supplementary Figs. 2-16**). When fine-tuning the model, CodonTransformer generated sequences with lower CSI than the base model for all organisms except *S. cerevisiae*, *N. tabacum* and its chloroplast which showed an increase in the CSI better mimicking the top 10% genomic CSI on which they were fine-tuned (Fig. 2, Supplementary Figs. 2-16). As expected, the clustering of CodonTransformer embeddings for organisms used for training resembles their overall phylogenetic distances (Supplementary Fig. 17).

3. In Figure 3, why only select 50 random genes? How did other methods generate sequences? In particular, how did the author select these 5 genes for demonstration in Figure 3b? And statistical testing results should be presented in Figure 3c.

In Fig. 3, we compared natural sequences to the ones generated by different tools from multiple DNA synthesis companies. Those tools are only accessible through a web interface therefore limiting the throughput. To acquire a representative sample while keeping the manual work manageable, we limited ourselves to 50 genes per organism. Tools offered by private companies are proprietary and underlying algorithms are not disclosed apart from commercial announcements. We also included in our benchmark an open-source deep learning model “ICOR” which works only for *E. coli*. The 5 genes illustrated in Fig. 3b were selected as representative around the DTW median of the 50 genes (since the genes are fixed it is not possible to select a single gene that well-aligns with all medians) and %MinMax profiles for all the other genes were added to Supplementary Data 1 (sheet %MinMax profiles). We modified the phrasing in L193-196 to further clarify this.

We implemented statistical testing in Fig. 3c according to the Reviewer’s comment (as shown below). We compared DTW distances of sequences generated by different models to the ones generated by fine-tuned CodonTransformer using unpaired t-test (the modified plot is shown below). We further discussed the statistical results in L214-217.

[L193-196]:

We first calculated the %MinMax profiles for all natural and codon-optimized sequences (the profile for one gene as representative of the Dynamic Time Warping (DTW) distance for each organism is shown in Fig. 3b, and for all genes in **Supplementary Data 1**)

[L214-217]:

The DTW distance is not significantly different between sequences generated by base and fine-tuned CodonTransformer, except for *E. coli* that the fine-tuned model showed a higher match to natural genes (i.e., lower DTW) (Fig. 3c). Fine-tuned CodonTransformer performed as well or better than other models except in *A. thaliana* where Twist DTW distances were significantly lower.

4. In section “Model benchmark with proteins of biotechnological interest”, the author compared sequences generated by different tools with each other rather than “ground truth”, which may be fine per

se but this referee would question that such result could be used to support author's claim "...CodonTransformer extrapolates natural-like features to new sequences" (line 245).

We changed the conclusion of this results section to "...corroborates observations made on natural sequences" (L282-283). In this section of the results, the sequences to optimize are foreign to the organisms and there are no natural sequences to use as "ground truth". We reformulated the introductory paragraph (L238-241) to clarify this point. The DTW distances shown in Fig. 4c for proteins of biotechnological interest, closely match the values found for natural genes in Fig. 3d. The fact that DTW distances as well as RNA folding free energy and GC content patterns observed for natural sequences hold true suggests that the models behave similarly when facing novel/heterologous sequences.

[L238-241]:

To further compare the performance of codon optimization tools in the context of heterologous expression, we collected 52 recombinant proteins with applications in molecular biology and therapeutics (**Supplementary Data 2**). Using each optimization tool, we designed DNA sequences for expression in *E. coli*, *S. cerevisiae*, *A. thaliana*, *M. musculus*, and *H. sapiens*. (**Supplementary Data 2**).

[L282-283]:

Altogether, these results suggest that heterologous sequences generated by codon optimization tools differ in their global and local codon usage and corroborate observations made on natural sequences.

5. The last but not the least, this referee would suggest a real-world case for demonstrating the superiority (and usage) of CodonTransformer.

We thank the Reviewer for this comment. A similar point was also raised by Reviewer 1 with a suggestion for predicting synonymous mutations with experimentally reported outcomes. In attempt to gather experimental data, we found a dataset of mutations in *ccdA*, *i.e.*, the antitoxin component of the *Escherichia coli* *ccdAB* toxin-antitoxin system, from a recent study (Chandra *et al.* 2022, *Molecular Biology and Evolution*, doi.org/10.1093/molbev/msac187). This data includes 62 synonymous mutations with their respective relative fitness and ribosome stalling values. We added a paragraph for the prediction results on this data and a new figure (2b) as follows:

[L167-177]:

Although we developed CodonTransformer for sequence design tasks, it can also be used to predict the effect of synonymous mutations. To do this, the probability of mutant and wild-type codons can be predicted by the model for every amino acid position. We used 62 synonymous mutations in *ccdA*, *i.e.*, the antitoxin component of the *E. coli ccdAB* toxin-antitoxin system, from a recent study³² with their respective experimental relative fitness and ribosome stalling values. Both base and fine-tune CodonTransformer models predicted the log-likelihood of the mutant codon significantly correlated with experimental relative fitness in a zero-shot manner with the wild-type DNA sequence as input to the model (Fig. 2b). The same prediction task by BFC that selects codons based on organismic codon usage table, also resulted in significant correlations slightly less than CodonTransformers. However, among BFC, base and fine-tuned CodonTransformer, only the fine-tuned model showed a significant correlation between log-likelihood of mutation and

relative ribosome stalling (Fig. 2b).

[L320-322]:

We then showed that base and fine-tuned models can predict the effect of synonymous mutations, providing an additional feature in relevant applications (Fig. 2b).

Reviewer #3 (Remarks to the Author):

The relevance of this subject is limited for a broad audience, but it is still an interesting and technically valuable contribution.

This paper presents CodonTransformer, a novel approach to codon optimization using a deep learning model based on Transformer architecture. The study results appear comparable to or better than previous approaches.

I appreciate the generalization of codon optimization across 164 organisms, covering a wide variety of species from different kingdoms of life, including both eukaryotes and prokaryotes. This diversity in training data is commendable and represents a substantial leap over single-species optimization approaches.

Response to Reviewer #3:

We thank the Reviewer for their assessment of our work and for raising key queries concerning the model bias, the organism embedding, and the inter- and intra-species variability of codon usage, which we address in the revision. Below, we provide point by point responses (in blue) and actions (in brown) with changes in the main text (underlined).

However, some aspects still need to be clarified. For example, does CodonTransformer focus primarily on subsets of codons with higher frequency, as the violin plots suggest (if I am not mistaken)? There appears to be a discrepancy between the spread of genomic data and the generated sequences—could this indicate a bias in the model toward certain codon choices? Clarifying this would strengthen the paper's claims about producing natural-like codon distributions.

The Violin plots in Fig. 2a represent CSI values measuring the similarity between codon usage in a sequence and the codon usage frequency table of a genome. The model re-designed DNA sequences for natural proteins with a higher CSI than genomic sequences, demonstrating that the model learned specific codon preferences for each organism (the aim of Fig. 2b). Obtaining

higher CSI than genomic sequences is expected because not every natural DNA sequence in a species fits the overall codon usage, whereas CodonTransformer optimizes every sequence based on codon frequencies that it encountered during training. We modified the relevant sections in the main text (L115-146) to clarify this point.

Regarding potential model bias, CodonTransformer preferentially selects codons with higher frequency as those were more encountered in the training dataset. We added a new supplementary figure (**Supplementary Fig. 24**, also shown below), representing the number of occurrences of each codon within the entire genome for *E. coli*, *S. cerevisiae*, *A. thaliana*, *M. musculus*, and *H. sapiens*, and within the corresponding sequences generated by base and fine-tuned CodonTransformer. Compared to genomic sequences, CodonTransformer amplifies differences between synonymous codons. Such behavior is expected, as CodonTransformer picks more frequently codons that it encountered during training. We mention this figure when discussing equivalent codon distributions for natural and benchmark genes (L261-269) and highlight the effect of fine-tuning on codon distribution in the discussion section (L339-341).

[L115-146]:

We trained the base model, which we named CodonTransformer, using ~1 million genes from 164 organisms (**Data availability**). The training set is a collection of genomes from all domains of life, i.e., Bacteria, Archaea, and Eukarya that constitute 56.1%, 2.5% and 41.4% of sequences, respectively. This model can be either used directly for codon optimization across species or it can be fine-tuned on custom sets of DNA sequences to perform more tailored tasks. In this study, we fine-tuned CodonTransformer on the 10% genes with the highest CSI (Methods) for 15 organisms: Escherichia coli, Bacillus subtilis, Pseudomonas putida, Thermococcus barophilus, Saccharomyces cerevisiae, Chlamydomonas reinhardtii and its chloroplast, Arabidopsis thaliana, Nicotiana tabacum and its chloroplast, Caenorhabditis elegans, Danio rerio, Drosophila melanogaster, Mus musculus, and Homo sapiens (Fig. 1c). The CSI²⁵ is derived from the codon adaptation index (CAI)⁸ to quantify the similarity of codon usage between a sequence and the codon usage frequency table of an organism instead of an arbitrary reference set of highly expressed genes. It therefore provides a robust metric at the multispecies level and may constitute a superior predictor of expression level in higher eukaryotes^{4,7,25,31}.

CodonTransformer learned codon usage across organisms

To assess the ability of the model to capture organism-specific codon preferences, we generated DNA sequences for all proteins encoded by the 15 genomes for which we performed fine-tuning (Fig. 1c). We then compared the sequences generated by the base and fine-tuned models to their natural counterparts (entire genome and top 10% CSI used for fine-tuning).

Sequences generated by the base model show a higher percentage of matching codons with their natural DNA counterparts than randomly selected codons without and with organismic preferences, uniform random choice (URC) and background frequency choice (BFC), respectively (Supplementary Fig. 1). Those sequences have a high CSI indicating that they follow for each organism the preference of codon usage (Fig. 2, Supplementary Figs. 2-16). Obtaining higher CSI than genomic sequences is expected because not every DNA sequence of a species is tuned for optimal expression, whereas CodonTransformer optimizes every sequence based on codon frequencies that it encounters during training. The base model generated sequences with higher CSI than the top 10% genomic CSI for all organisms except *S. cerevisiae*, *N. tabacum* and its

chloroplast (**Fig. 2, Supplementary Figs. 2-16**). When fine-tuning the model, CodonTransformer generated sequences with lower CSI than the base model for all organisms except *S. cerevisiae*, *N. tabacum* and its chloroplast which showed an increase in the CSI better mimicking the top 10% genomic CSI on which they were fine-tuned (Fig. 2, Supplementary Figs. 2-16). As expected, the clustering of CodonTransformer embeddings for organisms used for training resembles their overall phylogenetic distances (Supplementary Fig. 17).

[L261-269]:

To explain the differences between these two similarity measures, we looked at the codon distributions across the 52 sequences (Supplementary Fig. 22). For the 5 organisms, IDT and Twist generated sequences with more uniform codon distributions resulting in codons being present in a large number of sequences and explains the high Jaccard index between these models. The low sequence similarity suggests that the codons are positioned differently within the sequences. On the contrary, CodonTransformer and Genewiz displayed high differences in their total count of codons and number of sequences where a codon is present. These distinct model behaviors were also observed for the 50 natural sequences selected for each organism (Supplementary Fig. 23) which recapitulated distributions observed at the genome level (Supplementary Fig. 24).

[L339-341]:

Our results demonstrate that the fine-tuning process further allows modulating the model outcomes (**Fig. 2**), tuning the distribution of codons (Supplementary Figure 24), generating sequences with more natural-like patterns (**Fig. 3**) and reducing the number of negative cis-elements in *E. coli* (**Fig. 4d**).

Additionally, it would be helpful to know how organism-specificity is encoded. Is the organism specification introduced via one-hot encoding, or is there a more complex embedding scheme in use? Understanding this would provide better insight into how the model handles multispecies optimization.

The organism encoding was implemented using `token_type_id` that has been introduced into Transformers to allow them to model multi-protagonist discourse, like in questions and answers. As mentioned in the results section about the model (L100-106), `token_type_id` can be used to specify any type of context for string-like data. We, therefore, amplified the token type vocabulary so that every species has its own token type. This strategy allows our model to learn distinct codon preferences for each organism, associating specific codon usage patterns with their corresponding species. To clarify the strategy on this important aspect of our model, we added a new panel into the revised Fig. 1 (shown below) illustrating the organism encoding strategy used in CodonTransformer.

[L100-106]:

Token types are often used to distinguish interlocutors in text data (e.g., question vs answer or user vs assistant) but they can be used to specify any type of context for string-like data. We therefore amplified the token type vocabulary so that every species has its own token type (**Fig. 1b**). This strategy allows our model to learn distinct codon preferences for each organism, associating specific codon usage patterns with their corresponding species. In addition, passing the token type as an argument allows users to optimize a DNA sequence in the species of their choice.

Another important aspect to explore is the intra- and inter-species variability in codon usage. Does CodonTransformer accurately capture this variability, or does it tend to amplify or smooth out these differences? This point deserves further elaboration and validation within the study.

We do not differentiate between subspecies/strain sequences during training. For sequences within the same organism, CodonTransformer generates less diverse (high) codon adaptation index (CSI) values. This is expected because not every DNA sequence of a species is tuned for optimal expression, whereas CodonTransformer optimizes every sequence (please see our response and action to the first comment).

For inter-species variability, we expect that CodonTransformer captures phylogenetic distance as it is reflected in the codon usage preferences (doi.org/10.1016/j.ympev.2019.106697). To address this comment, we added Supplementary Fig. 17 representing CodonTransformer embedding for organisms used for training (all 164 species) and fine-tuning. These embeddings were plotted using hierarchical clustering which groups similar embeddings into clusters based on their variance, creating a hierarchical tree structure. The resulting dendrogram visually represents these clusters, showing the relationships and distances between the embeddings of different organisms. For example, eukaryotes are clustered together with evolutionary close organisms being the closests.

[L145-146]:

As expected, the clustering of CodonTransformer embeddings for organisms used for training resembles their overall phylogenetic distances (Supplementary Fig. 17).

Reviewer #2 (Remarks to the Author):

This referee would thank for the authors' extensive works which have effectively addressed previous concerns. Now this referee believes the manuscript is ready to move forward.

The comments below are based on personal understanding of Reviewer 1's comments and author's responses:

The authors have generally addressed almost all comments from Reviewer 1. Specifically, I still find the last comment from Reviewer 1 (-The ultimate question in codon optimization is not whether we can recover ...) was not well-explored. I understand the difficulty in wet-experiment, so I think this's acceptable so far.

Reviewer #2 (Remarks on code availability):

The GitHub repo is overall well-organized and actively maintained.

Reviewer #3 (Remarks to the Author):

All my issues have been addressed. I have no more points to raise.

Reviewer #4 (Remarks to the Author):

Response to the Reviewers

Once again, we would like to thank all the reviewers for their constructive comments and their help in improving our manuscript. We are delighted to see that our revised manuscript addressed the Reviewers' concerns.

As pointed out by Reviewer #2, we used quality data for synonymous mutations, as suggested by Reviewer #1, and calculated the log-likelihood of mutant codons over wild-type codons. Results showed that CodonTransformer is able to capture the effect of synonymous mutations in this dataset.